# An Empirical Study of the Restoration Potential of Urban Deciduous Forest Space to Youth

**DOI:** 10.3390/ijerph19063453

**Published:** 2022-03-15

**Authors:** Linjia Wu, Qidi Dong, Shixian Luo, Yanling Li, Yuzhou Liu, Jiani Li, Zhixian Zhu, Mingliang He, Yuhang Luo, Qibing Chen

**Affiliations:** 1College of Landscape Architecture, Sichuan Agricultural University, Chengdu 611130, China; wulinjia0716@163.com (L.W.); dongdd0805@163.com (Q.D.); lyz18382424843@163.com (Y.L.); zaq135@126.com (J.L.); hemingliang@stu.sicau.edu.cn (M.H.); 2020210007@stu.sicau.edu.cn (Y.L.); 2Graduate School of Horticulture, Chiba University, Chiba 263-8522, Japan; 19hd4106@student.gs.chiba-u.jp; 3School of Literature and Arts, Southwest University of Science and Technology, Mianyang 621010, China; yanling-li0611@hotmail.com; 4School of Fine Arts and Design, Chengdu University, Chengdu 610106, China; zhuzhixian@cdu.edu.cn

**Keywords:** urban forests, deciduous forest, ginkgo, forest space, physiological–psychological response, preference

## Abstract

Urban deciduous forests are an important ecological resource and seasonal landscape in the urban environment. However, in the abundant literature on how urban green space promotes human health and well-being, research on urban seasonal deciduous forests is limited. This study aimed to investigate the physiological and psychological recovery potential provided of urban deciduous forest space for youths and the spatial preferences of youths regarding such spaces. We recruited 120 participants to study the restorative potential of two typical urban deciduous forest landscape spaces (experimental groups) and one urban road environment (control group). The results showed that after 15 min of observation, the blood pressure (especially the diastolic blood pressure (*p* < 0.01)) and pulse of the deciduous forest trail setting (DFTS) group effectively decreased, and the restorative mood significantly increased. Regarding change in emotional parameters, the DFTS group scored higher on “interest” and significantly higher than the other two groups on positive emotion. The correlation results show that density and level are the key factors affecting spatial preferences regarding complex deciduous forests. An increase in density reduces the mood of re-laxation, and an increase in level decreases fatigue and interest. We suggest (1) constructing foot-paths in urban deciduous forests to reduce their spatial density as to improve the relaxation effect and (2) increasing landscape diversity according to the forest space to facilitate user participation and interest. This study provides a scientific basis for the environmental restoration of deciduous landscapes and for urban forestry management decision-makers based on space type construction.

## 1. Introduction

Urban forests are an important natural resource in urban areas. In the process of globalization and urbanization, as a restorative environment, urban forests is of great significance to reducing the economic burden of medical expenses [1,2]. Actively constructing and managing urban forests as a means to address climate change and support community health has gradually become regarded as a potentially beneficial solution for urban society and has been widely pursued [3]. Promoting the development of urban forests is conducive to ecosystems and human health [4], and an increasing number of scattered forests are being included in landscape protection plans [5]. In the research on how urban forests promote human well-being, recovery is defined as a series of processes leading to the renewal of physiological and psychological resources [6], whereby the most prominent components of natural recovery include attention [7] and physiological and psychological stress recovery [8]. Many researchers have ascribed the health benefits of urban forests for the human body to several biological mechanisms. For example, the biological sound effects created by various spatial structures [9] in urban forests have been positively correlated with pleasure perception [10]. The differential performance of the characteristic combination of elements in the forest is reflected in the restoration potential [11,12,13]. Urban forests have been viewed as “community living rooms” that can provide venues for public participation in social activities that promote happiness and social health [14]. Although the cited studies reveal the health benefits of urban forests, there has been little discussions on the differences in health promotion effect and preferences regarding the spatial constructions of seasonal urban forests. Urban public health and forestry management require a more detailed basis for spatial optimization.

Urban forests provide a large number of ecological services for cities, including the urban environment and urban society, improve human well-being [15,16]; and are an important concept in the area of new forestry types [17]. A Helsinki social value mapping study showed that the most valuable green spaces are usually relatively large forested areas [18], including urban forests. This green space, which represents a close-to-hand from of “wild nature”, is considered to play a more important ecological role than most other green spaces within cities [19], and it provides a large number of ecological services for cities, including for the urban environment and urban society, while improving human well-being [20]. However, historically, urban afforestation, especially in developing countries, has been viewed using aesthetic, financial, and environmental capacity as forestry criteria [21,22]. The research data based on the positive impact of urban forests on health can be used as an important basis for guiding forest management [16].

Previous studies have confirmed that the positive physiopsychological feedback from characteristic stimuli [23] varies across different green environment types in cities. From the perspective of theory and experience, this phenomenon is associated with dual perception, i.e., multisensory physical experience and a personal perception track [24], and includes a range of changes caused by the things people prefer [25]. In efforts to promote urban human health, urban planners have used forest therapy [26,27] to benefit urban residents, but studies on urban forests environments have mostly focused on simple comparisons of physiology and psychology within a built environment and the restorative characteristics exhibited by different forest types [28], forest distances [29], and forest management systems [30]. For example, for nearby residents, the overall restoration intuition is higher in roughly managed woodlands than in large parks [31]; people feel more “pleasure” in a nurtured forest [32]; and open, unobstructed forests enhance positive emotions [23,33]. In seasonal green space restoration research, previous studies reflect to a limited degree the different restorative potentials of different seasonal landscapes [34], and only a few scholars have conducted controlled experiments in environments without green space [35] or compared the restorative effects of the same environment in different seasons [34]. More specific research on the restorative potential of and preferences for different landscape types in built up urban deciduous forest environments to guide design and planning remains limited.

*Ginkgo biloba* is an important ornamental and medicinal tree endemic to China that is widely planted in East Asia, Europe and the United States, and can be found everywhere in urban reforestation areas. *Ginkgo* leaves represent a “link” between plants and the soil in autumn [36,37] and have the characteristics of creating a large amount of litter, fast decomposition, good nutrient return, and good soil fertility maintenance [38]. As a landscape tree species, seasonal *Ginkgo biloba* creates a good urban landscape. There are few studies on the restorative potential of deciduous forests; however, the use of deciduous broadleaf forest landscape resources as restorative material has important research value [28].

In this paper, the deciduous environment of a *ginkgo biloba* forest in autumn was chosen as the study object for two reasons. First, *ginkgo* is one of the most widely planted deciduous trees in temperate regions and an important landscape tree species. The yellow color of the tree’s leaves in autumn is representative of the deciduous landscape and is universal and typical [38,39]. Second, in Chengdu, Sichuan Province, urban forests construction planning was initiated early, with the city reaching 36.15% forest coverage and taking the lead in implementing deciduous landscape protection measures in 2009. These circumstances gives our research practical significance. Therefore, our study focuses on a common behavior in urban forests environments, i.e., viewing activities [40], as a form of interaction between users and the deciduous forest environment as well as the urban environment. This study tested the hypothesis that a short period in a deciduous forest interior setting (DFIS) or a deciduous forest trail setting (DFTS) could cause participants to relax physically and mentally. We established a typical city setting (CS) as a reference, one in which common urban elements could be observed, and differences in the participants’ physical and psychological relaxation and preferences were investigated. This study provides a reference for future forest construction for urban greening management decisions and urban planning.

We have three hypotheses as follows. Compared with that of the urban environment, (1) the stimulation of two types of deciduous forests can generate positive physiological and psychological feedback [41,42], (2) the DFTS will improve participants’ physical and mental states more significantly, and (3) participants’ preference for the DFIS will be greater.

## 2. Materials and Methods

### 2.1. Participants

We recruited 120 students (mean age: 21.80 ± 2.14 years; mean body mass index: 21.20 ± 3.63; 60 males and 60 females) from Sichuan Agricultural University through posters and school social platforms. Study participants were required to have no personal history of physical or mental illness. We randomly divided the participants into three groups with approximately even gender distribution. According to stimulus setting, we registered the DFIS, DFTS, and CS group volunteers with volunteer numbers SA1-40, SB1-40, and CS1-40, respectively. Because the experiment involved brain wave detection, to exclude cerebral hemispheric differences, all participants were right-handed [41]. *Ginkgo* is a plant commonly used for greening in China, especially in the southwestern region, where this experiment was conducted. Therefore, experimental interference from unfamiliarity with the tree species was excluded. Demographic information such as the gender and age of the participants, was collected through a questionnaire prior to the experiment. The participants were informed in detail regarding the experimental procedures and voluntarily signed an informed consent form. The volunteers were instructed to avoid drinking, smoking, and strenuous physical activity before participating in the experiment. The experimental study procedures were conducted in accordance with the ethical standards of the National Research Council and in compliance with the Helsinki Declaration. They were also approved by the local ethics committee of the School of Landscape Architecture, Sichuan Agricultural University.

### 2.2. Study Sites and Procedures

The experiment was conducted in the largest ginkgo garden and adjacent neighborhood open space in Chengdu, Sichuan Province (103°86′44″ E, 30°58′50″ N). The *Ginkgo biloba* garden covers an area of more than 8000 m^2^, in which more than 100 *Ginkgo biloba* of different tree ages are planted, with an average diameter at breast height of 125 cm. The garden is an approximately 20 min drive from the city center. The canopy density in the forest is 0.45, i.e., a moderate canopy forests. The garden is an urban autumn forest recreational site. Visitors, parking lots, trails, and other typical urban garden elements can be found in the park. To reduce the confounding effects caused by plant density and parallel experiments, as the DFIS experimental site, we selected an area with a relatively more uniform forest stand [43] and away from trails, s, i.e., an 84 × 72 m section of pure Ginkgo biloba forest space. For the DFTS viewing test point, path with Ginkgo biloba on both sides and a general footpath width, i.e., 1.5–3 m, were selected as candidates for the experimental site. Finally, a U-shaped trail with an average width of 3.2 m and a total length of 166 m was selected. This path was convenient for viewing *Ginkgo biloba*. The vegetation structure of the two experimental sites was an arbor grass structure, whereby the arbor consisted of *Ginkgo biloba*, and the ground cover was a mixed ground cover consisting of *Cynodon dactylon (L.) Pers.* and *Poa annua L*. The control sample site was close to the two experimental stimulation sites and consisted of a 46 × 24 m^2^ block open space at an urban T-intersection (Figure 1).

The experiment was divided into three phases (Figure 2). First, all participants took a pretest in a classroom at Sichuan Agricultural University, which included questionnaire completions and physiological measurements (classroom temperature 22–26 °C and classroom humidity 49–55%). The questionnaire included key sociodemographic characteristics and two psychological state scales (Profile of Mood States, POMS, and Restorative Outcome Scale, ROS), and the physiological measurements included blood pressure, blood oxygen, and pulse rate. The participants then traveled to the experimental sites (approximately 30 min from the school) in a university vehicle. When the participants arrived, they sat quietly for 5 min to eliminate external influences [44,45] before being led by staff on a walk to the two deciduous forest environments and the control environment to begin a simultaneous 15-min viewing activity. An EmotivPRO device was used to simultaneously record participant electroencephalogram (EEG) data. The volunteers were asked to experience the environment quietly, without using cell phones, talking, eating or drinking. After 15 min, the EEG instrument was removed and physiological and psychological measurements were repeated. Subsequently, preference and spatial perception questionnaires were completed, and the volunteers were brought back to the university by staff, ending the experiment. The entire experiment was completed over three days (3–5 November 2020), starting each day at 9:00 a.m. The environmental factors of the two experimental and control groups were measured every 2 h using a black sphere thermometer (AZ8778), a noise meter (CEN-TER322), and a digital photometer (T-1H, Minolta, Osaka, Japan) (Table 1).

### 2.3. Measure

#### 2.3.1. Blood Pressure, Blood Oxygen and Pulse Measurement

Blood pressure, including systolic (mmHg) and diastolic (mmHg) pressure, and pulse rate (bmp) were measured using a sphygmomanometer (Omron, HEM-6322T, Tokyo, Japan). Systolic and diastolic blood pressure increase when a person is stressed and decrease when he or she is relaxed, while pulse rate increases when the body is in motion or emotionally excited. These measures represent a common method to assess the effect of forest therapy [46,47,48].

#### 2.3.2. Measurement of Neuroaffective Parameters

EEG is used to record changes in electrical waves during brain activity [49]. It is recorded using neuroharmonic EEG biofeedback techniques [50,51] and widely used as emotional feedback in restorative research [52]; it is noninvasive to humans. The EEG recorder Emotiv EPOC+ device used in this study is a noninvasive EEG signal acquisition instrument with an internal signal sampling frequency of 1024 Hz and an internal sampling frequency of 128 Hz per channel; the instrument is worn so as to cover four brain lobe regions (i.e., the frontal, temporal, parietal, and occipital lobes) using 14 channels (AF4, AF3, F3, F4, F7, F8, FC5, FC6, T7, T8, P7, P8, O1, and O2) for recording [39]. After a subject dons the device, the output of the four brain lobe regions with respect to six emotional (“engagement”, “excitement”, “stress”, “relaxation”, “interest” and “focus”) is collected. These parameters objectively reflect the emotional feedback of the subject in response to different environmental stimuli [53] (Figure 3).

#### 2.3.3. POMS Emotion Measurement

POMS is a reliable and valid psychometric instrument that includes 40 adjectives, rated on a scale of 0–4 (0 = none; 4 = very strong), which can be integrated into seven valid dimensions: tension and anxiety (T-A), depression (D), anger and hostility (A–H), vitality (V), fatigue (F), confusion (C), and self-esteem (S). Three psychological indicators are also available: positive, negative, and total mood (TMD) [44].

#### 2.3.4. Restorative Emotion Measurement

Restorative emotion was measured using the ROS. The ROS [54,55] is a reliable scale for measuring the effects of restorative forest environments [56]; it contains six items (“I feel restored and relaxed”, “I feel calm”, “I feel I have enthusiasm and energy for daily life”, “I feel focused and alert”, “I can forget my daily worries”, and “My thoughts are clear”). Each item is measured on a 7-point Likert scale (1 = not at all; 7 = extreme).

#### 2.3.5. Preference and Spatial Scale Measurement

The participant’s preference regarding their deciduous environment was determined by a five-point Likert scale, and spatial perceptions regarding deciduous landscape type were recorded using a semistructured questionnaire that included spatial density, height, level, and overall [57]. Scores for each perception of space ranged from 0 to 4 (0 = not at all; 4 = very much) (Table 2).

### 2.4. Statistical Analysis

Statistical and data analyses were performed using Excel 2016 and SPSS 20.0 (IBM Corp., Armonk, NY, USA). Descriptive statistics and chi-square tests were used to examine the sociodemographic characteristics of the study participants (Appendix A), and paired t-test were used to compare the mean physiological parameters between the two sites. Wilcoxon signed-rank tests were used to analyze differences in psychological indicators after viewing in the two settings, and one-way ANOVAs were used to examine the effect of experimental site differences on participant preferences.

## 3. Results

### 3.1. Blood Oxygen, Blood Pressure, and Pulse Rate

As shown in Figure 4, there were no significant overall changes in the blood oxygen indicators after 15 min of viewing activity in the three environments. Participants in the DFTS exhibited significantly decreased systolic blood pressure (115.35 ± 12.07 before and 103.45 ± 13.47 after viewing; *p* < 0.01), diastolic blood pressure (76.66 ± 8.11 before viewing and 67.10 ± 7.83 after viewing; *p* < 0.01), and pulse (83.15 ± 16.91 before viewing and 74.52 ± 10.10 after viewing; *p* < 0.05). Systolic blood pressure in the DFIS group was significantly decreased (118.7 ± 12.75 before and 109.55 ± 11.90 after viewing; *p* < 0.05). The blood pressure and pulse indices of the control group showed an increasing trend before and after the experiment, but there was no significant change before and after.

### 3.2. Neuroemotional Parameters

Figure 5 shows the differences in the mean values of the neuroemotional parameters between the groups after 15 min of viewing in the different environments. Compared with the control group, the six indices of affective parameters in the DFTS group were significantly different, and there were significant differences in “engagement”, “stress”, and “interest” between the DFIS group and the control group. Regarding the two experimental groups, the DFIS group scored significantly higher than the DFTS group on the indicators of “engagement”, “stress”, “excitement”, and “interest”. “Focus” was significantly lower in the DFIS group compared to the DFTS group, and the “Relaxation” indicator was not significantly different. Figure 6 shows the minute-by-minute changes in the six emotional parameters during the viewing period. One minute after the start of measurement, all three groups displayed significant changes in their emotional parameters. In the CS group, the values of “engagement”, “excitement”, and “stress” were higher than those of the two experimental groups, while the values of “relaxation”, “interest”, and “focus” were lower than those of the two experimental groups, which is consistent with the results of the comparison between groups. The fluctuation of the EEG emotion parameters in the DFTS group and CS group was greater than in the DFIS group.

### 3.3. POMS

According to our results (Figure 7 and Figure 8), after viewing the three environments, the DFIS group exhibited a significant decrease in “confusion” emotions (0.88 ± 0.68 previewing and 0.47 ± 0.51 postviewing; *p* < 0.01). In contrast, the DFTS group showed a significant increase in positive emotions compared to the other two groups, the “vigor” and “self-esteem” (*p* < 0.01) were increased, and the negative emotions “tension and anxiety” (*p* < 0.01), “anger and hostility” (*p* < 0.05), and “fatigue” (*p* < 0.01)) were reduced. According to the calculation of seven emotional indicators, three emotional state indicators were obtained. The DFIS group had significantly higher positive (M = 20.25 ± 7.89) (“M” means “mean”) and TMD (M = 90.95 ± 15.16) (“TMD” means “total mood”) emotional values than the CS group, and the DFTS group had higher positive (*p* < 0.01) and TMD (*p* < 0.05) emotional values than the CS group. Regarding negative emotion values, the DFIS and DFTS groups scored significantly lower than the CS group (DFIS: M = 11.20 ± 10.61; DFTS: M = 11.20 ± 9.00; CS: M = 18.65 ± 13.69). The DFTS group scored significantly higher than the DFIS group in positive emotions, and the group’s negative emotions and TMD did not significantly differ from those of the DFIS group.

### 3.4. Restorative Emotions

The restorative effects on the emotions of the two type of deciduous environments and the urban environments were measured with the ROS (Table 3). The difference between the two deciduous environments before and after viewing was significant (*p* < 0.05). The restorative effect of DFTS (4.60 ± 1.00 previewing and 5.31 ± 0.87 postviewing, *p* < 0.01) was highly significant. However, there was no significant difference between the two deciduous environments. The restorative effect of the urban environment was not significant.

### 3.5. Preference and Correlation Research

To assess deciduous environment preferences, only the DFIS and DFTS groups were scored. When the two groups of preference data were relatively independent and met normal distribution requirements, we tested the homogeneity of variance (*p* = 0.176 > 0.05), accepted the original hypothesis, and performed one-way ANOVA with the grouped volunteers’ preference values as the dependent variable and the two types of deciduous environments as independent variables, followed by post hoc tests. The DFIS and DFTS groups (M = 3.6, SD = 0.82) scored similarly, whereby the DFTS group (M = 4.00, SD = 0.73) scored slightly higher, but the difference between the environmental preferences of the two groups was not significant (*p* = 0.111).

In investigating the relationship between deciduous landscape space and health benefits (Table 4), we found that the four spatial indicators were not significantly correlated with each of the physiological indicators. Among the spatial indicators and mood correlations, “density” was significantly negatively correlated with “relaxation” (−0.413 **, *p* < 0.01) and “interest” (−0.385 *, *p* < 0.05); “height” was significantly negatively correlated with “relaxation”; “level” was significantly negatively correlated with “relaxation” (−0.43 **, *p* < 0.01) and “interest” (−0.414 **, *p* < 0.01) and was negatively correlated with “focus” (−0.314 *, *p* < 0.05) and significantly positively correlated with “engagement” (0.563 **, *p* < 0.01); and “overall” was significantly positively correlated with “positive mood” (0.005, *p* < 0.05) but not with other emotions.

## 4. Discussion

### 4.1. Effect on Blood Pressure, Blood Oxygen and Pulse

It is known that youth populations are more vulnerable to various stressors, such as academic, social, and employment stressors [58,59]; therefore, this empirical research may be valuable for improving stress coping and providing other health benefits for urban youth populations. Previous studies have shown that people can experience physical and mental relaxation in natural environments [60,61,62]. In an urban environment that is lacking in nature, urban plots with large trees and natural landscapes can be viewed as more coherent positive emotional environments [41]; with which to achieve the same recovery rate as can be achieved in a natural environment [63]. The results of this study, showed that diastolic blood pressure, systolic blood pressure and pulse rate decreased to different degrees in both deciduous forest environments groups compared to the control group. These findings confirm our first hypothesis that the seasonal deciduous forest environment in the city has a relaxing effect. However, there were index differences between the two deciduous environments. It was found that both deciduous forest groups experienced positive physiological effects compared to the control group. However the DFIS group did not perform as well as the DFTS group in terms of diastolic blood pressure, systolic blood pressure, and pulse rate reduction (Figure 4). This finding is inconsistent with our second hypothesis that the deciduous forest trail space would more significantly improve the physiological status of the participants. Diastolic blood pressure was significantly lower (*p* < 0.05) in both experimental groups after 15 min of observation (Figure 4), indicating enhanced parasympathetic activity and decreased sympathetic activity under the deciduous environmental intervention, with no significant changes before and after in the control group, which is consistent with Chorong et al. [33]. As this experiment involved a short period of environmental stimulation, blood oxygen was an important parameter of the respiratory cycle, and although studies have demonstrated that respiration is negatively correlated with canopy cover and leaf area index in forest structure [64], the experimental environment did not involve changes in the respiratory environment, and the blood oxygen index did not change significantly and was at normal values in both the experimental and control groups. This outcome is consistent with the findings of Lyu [65] and Zeng [40] in a three day bamboo forest environmental study.

In addition, several field trials have shown that natural environments relieve stress and reduce pulse and stress symptoms [47]. For the pulse index, compared to the urban environment group, the DFTS group showed a significant decrease in pulse index (*p* < 0.01) and the DFIS group (*p* < 0.01) showed no significant change in pulse, consistent with certain previous studies [66,67] and excluding participant differences between individuals and seasons [33]. Deng et al. found that the magnitude of pulse reduction varied across landscape elements, with topographic landscapes being more effective at inducing relaxation than turf and water landscapes [68], and recoverability as a component of the landscape [69] varies with the environment. The spatial composition outcomes of the two experimental groups differed when combined with the results for physiological indicators. We conjecture that the experience of the DFTS group had a more pronounced role in stimulating the sympathetic and parasympathetic nervous systems, which in turn affected participant’s mood. This difference is similar to the differences in the results of the present psychological response, but the more detailed indicators of heart rate variability and skin electricity associated with these differences require further study.

### 4.2. Effect on Neuroaffective Parameters

First, according to the EEG results (Figure 5), for the three groups of neuroemotional parameters, the control group showed higher levels of “engagement”, “excitement”, and “stress” than the experimental group. The DFIS group showed two higher emotional parameters: “engagement”, “stress” (*p* < 0.01). “Engagement” is a high arousal response in the classical model of arousal emotions [39], and “stress” can be viewed as a combined result of psychological and emotional value responses [25]. The DFIS group scored significantly lower than the control group for these three parameters, again confirming the first hypothesis of the study. Studies have shown that relaxation under high biodiversity conditions is in fact equivalent to low arousal and that dense plant landscapes are more conducive to health [70,71]. Although biodiversity was not used as an indicator in this study, a significant correlation was found between the spatial indicators “density” and “relaxation”, with no significant difference in plant density between the two deciduous environments used in the study, which could explain why the two experimental groups did not significantly differ in stimulation. Second, from a restorative perspective, interaction with nature improves cognition and attention [7,72]; the values of the “focus” neural parameter were significantly higher in the DFTS group than in the other two groups, suggesting that the DFTS group interacted participants more and were more likely to focus and improve their attention as they relaxed, which provides ideas for creating a meditation environment in urban green space.

The factors that affect neural activity are diverse, and in the minute-by-minute changes in EEG display (Figure 6), both the experimental and control groups produced different degrees of change one minute after environmental stimulation. This outcome is consistent with the findings of Wang Yu Xi et al. [49], who observed responses in subjects’ EEG one minute after they viewed a nature video; several studies have found that changes in EEG waves reach stability within five minutes regardless whether live or video stimulation is used [49], but in our study of minute-by-minute changes in EEG parameters, there was a manifestation of unstable changes in each emotional parameter after five minutes, which may be related to the interference of other factors in real-word stimulations. However, this paper is reluctant to draw conclusive inferences from the limited data available, and more research is required to determine the causes of these discrepancies.

### 4.3. Influence on POMS Emotion and Restorative Emotion

The study results showed that the two experimental groups had larger positive mood increases and negative mood decreases than the control group, and in terms of restorative mood, the experimental group had more “calm”, “enthusiasm”, and “focus”. The DFTS group scored higher than the DFIS group in positive emotions and restorative emotions. However, there was no significant difference between the two groups (Figure 7). This finding is consistent with Lin Wei’s [57] study on the superior psychological relaxation effect of forest path space over forest interior space in bamboo forest micro spaces, but the two experimental settings of this study were not nonurban area but rather seasonal deciduous forest in an urban area. As the nature-related landscape component was positively correlated with restorative scores [61,73], we assumed that positive emotions would be more pronounced in the interior space of deciduous forests with a high natural component, but apparently the results were the opposite. Involuntary attention to environmental stimuli in the two experimental groups produced different emotional expressions from an evolutionary psychological perspective, creating more emotions containing calmness, relaxation, and security in the DFTS group. Excluding the variability of the measured environmental indicators, we believe that the sense of place may be one reasons for this result. People have an ability to perceive space, and the spatial element of this spatial sense can make us physically and mentally happy [74,75], an outcome is similar to that of those with a degree of familiarity with nature and previous experiences affecting environmental restoration potential [76]. Additionally, a certain degree of individual attachment to place affects restoration levels [77]. Compared with the pure deciduous forest space, the deciduous forest trail space had a familiar pedestrian trail. As a place where urban residents often rest and play, the trail provided a higher sense of familiarity and interaction, thus producing a stronger relaxing effect. Therefore, this finding refutes the second hypothesis of this study. According to the relationship between existing forest environmental characteristics and health restoration, we can obtain a stronger sense of relaxation and positive emotion in an environment with rich biodiversity [57], openness [23,32], and low degree of closure [57]. The difference between the DFTS and DFIS is the presence or absence of the trail element. No plants can be planted on the trail, and the planting density of the DFTS is less than that of DFIS in a certain range, which makes the space open. Therefore, the DFIS scores for positive emotion were significantly lower than those of the DFTS group, and the DFIS group’s pretest and posttest scores for restorative emotion were not as significant as those of the DFTS (Figure 8, Table 3).

### 4.4. Influence on Youths’ Preference and Correlation

Different types of spaces trigger different physical and mental feedback, and environmental preference and liking are important manifestations of this feedback [39,56]. Based on the large number of studies demonstrating a high preference for natural landscapes compared to urban landscapes, we expected the DFIS with more natural elements to stimulate a higher preference. However, the DFTS was more popular. Although the high preference for the DFTS in this study is comparable to the restorative benefits results, it is worth noting that environment-specific preferences do not imply that an environment is highly restorative [28]. Based on Lin, W et al. [57], who found that “spatial diversity” and “closure” may be key structural factors influencing physiological and psychological stress in pure forest space, we extracted some common information from the preference indicator questionnaire, i.e., that the DFTS group scored higher for the “density” and “level” indicators. We found that the indicators of relaxation and interest decreased significantly when density and level increased to a certain level (Table 4). In three-dimensional space, density can remind participants of internal spatial chaos and depression, and level can represent vertical structure [57]. When the value of both increases, the spatial visibility of the landscape decreases and the complexity of the landscape increases, which leads to a decrease in fatigue and interest. Under similar conditions, the DFTS group could see more open space and had lower “closure” during observation, and under the premise of establishing a positive association between preference and recovery potential [78,79], we found that the DFTS group showed better recovery mood, a finding consistent with the preference results but contrary to the third hypothesis of this paper. Therefore, we suggest that “suitable planting density” and “suitable level” are the key factors influencing landscape preference in urban deciduous broadleaf forests.

### 4.5. Suggestions for Urban Deciduous Forest Construction Planning

As a complex dynamic system, urban forests management is particularly important [80]. It is urgent that such green public open space be available during pandemics, such as the COVID-19 pandemic [81]. Increasing urban intervention of such green space can better promote sustainable and healthy development [82], including ecological development under the global climate crisis [14], but green health promotion based on restorative potential may constitute a key urban health strategy that remains neglected by policy makers. Data from our study suggest that urban deciduous environments under seasonal change have restorative potential and urban deciduous forests with trails are better at promoting physical and mental relaxation and positive emotional benefits. Landscape designers seeking to plan and construct attractive forest space can support the fair use of youth groups in urban space. The inner space of a deciduous forest was more conducive to participants’ concentration, participation and interest. This finding indicates the emotional awakening value of recreational activities undertaken in pure forest. Although studies have shown the public’s preference for forest restoration of natural sites [83], diversified landscape structure contributes to the development of high willingness to pay [84]. Our research results provide data support for the health benefits created by urban forests trail space. Although we do not know whether the trail material will affect the health benefits of viewing under forest cover, as a natural place for outdoor entertainment, it is worth advocating improving the accessibility of urban forests through trail construction and increasing interaction to achieve a relaxation effect. In addition, the inner space of deciduous forests is more conducive to visitor concentration and increasing the visitor’s sense of participation and interest, which indicates the emotional arousal value of recreational activities in pure forests.

Many studies on human well-being have provided urban planners with recommendations for urban forests at different scales; on the small and medium scales, increased natural landscapes and mature trees in urban woodlands could improve the well-being of middle-aged women, increase biodiversity and landscape complexity [30], and maintain the expected benefits of trees in cities [80] while generating high aesthetic value, etc. Landscape designers planning and constructing attractive forest space can use such space to promote the fair use of youth groups in urban space. In the forest, participants pay attention to the interesting elements in the space [85], and composition characteristics can affect the experience of the landscape [86]. Urban forests planning can consider adding attractive visual elements such as colorful flowers and water [87] to improve spatial interest and participation. This study has revealed the key impact of “density” and “level” on spatial preference. Density is usually associated with the impression of safety [88]. Carefully considering the planting of deciduous mixed forest in forest planning can reduce the density in seasonal changes. When planning a footpath under a forest canopy, open spaces, such as viewing platforms, rest pavilions, and corridors, can be considered to reduce the degree of closure. Regarding the vertical structure of urban forests, an appropriate vegetation level is key. One should try to avoid creating visual fatigue and increase the spatial density caused by more vegetation levels, while reasonably plan the planning the planting density according to the forest level to create a better restorative or participatory environment.

### 4.6. Limitations

This study has several limitations. First, the importance of different groups in studies on restorative preferences and potential has often been emphasized. The participants in this study were youths, and whether they were from rural or urban areas may have been a confounding factor in the results. The findings are only representative of urban youth, and future studies should include a broader group. Second, while our study used parallel research to avoid legacy effects in all three settings, the rich diversity of urban landscape types, including topography, water, and other landscape elements, may potentially result in differing health benefits [68]. In future studies, we will investigate the spatial scale differences of deciduous forests landscape types in more detail so as to better guide urban deciduous forest landscaping. Finally, this study was a cross-sectional investigation, and seasonal differences in short-duration environmental conditions may lead to different restorative potentials, which may in turn be related to climate, environment, leaf color, and quantity. More research is required to determine the effects of different times on landscape restoration and preferences for deciduous forests.

## 5. Conclusions

The construction of urban forests environments is an important area of concern for urban planners and policy makers, and the health benefits of such environments require additional study and careful testing. This study on the differences in restorative benefits of and preferences regarding different spaces in urban deciduous forests arrived at three main conclusions. First, the study provides empirical evidence that seasonal deciduous forests in urban forests have a beneficial effect on youths’ physical and mental recovery. Second, after considering two different spatial plans for urban deciduous forests, the study found that deciduous spaces with trails have more positive affective and restorative benefits. Third, the “density” and “level” of urban deciduous forest are the key factors affecting spatial preference. To improve the urban population’s well-being, planners should optimize urban afforestation through a reasonable spatial layout.

## Figures and Tables

**Figure 1 ijerph-19-03453-f001:**
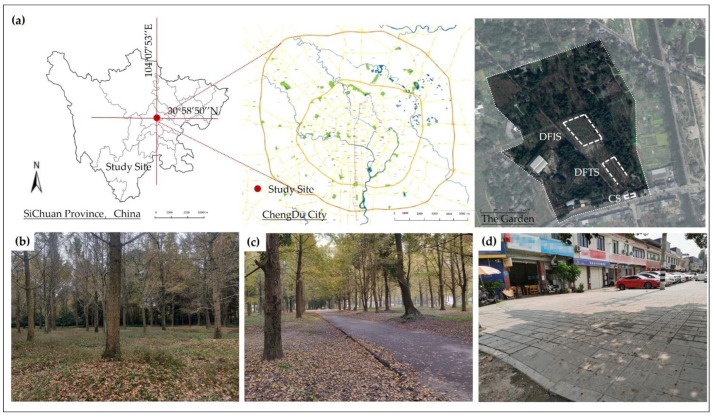
Study location and three stimulating environments: (**a**) map of the experimental site and study site; (**b**) DFIS: deciduous forest interior setting; (**c**) DFTS: deciduous forest trail setting; (**d**) CS: city setting.

**Figure 2 ijerph-19-03453-f002:**
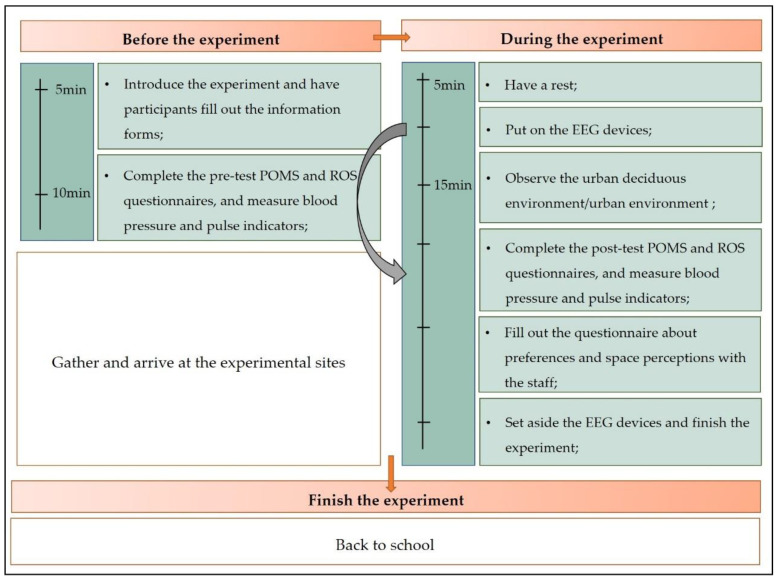
Experiment procedure (POMS = Profile of Mood States, ROS = Restorative Outcome Scale States, EEG = electroencephalogram).

**Figure 3 ijerph-19-03453-f003:**
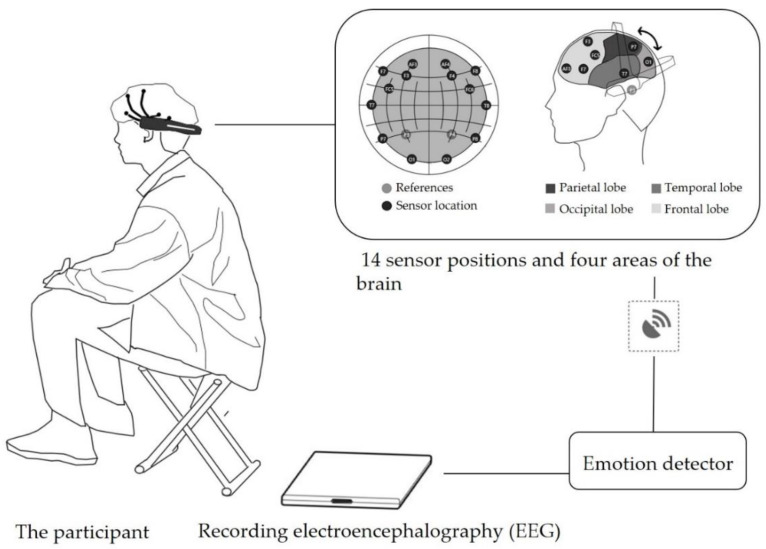
Participant wearing Emotiv EPOC + and the four covered brain regions.

**Figure 4 ijerph-19-03453-f004:**
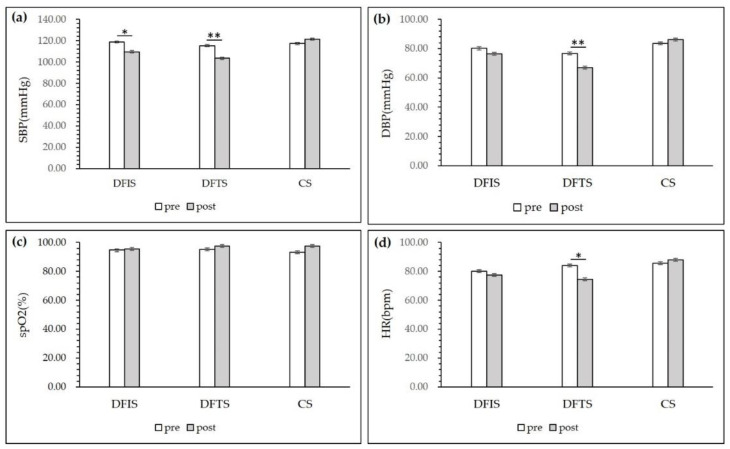
Systolic blood pressure, diastolic blood pressure, peripheral oxygen saturation, and heart rate of the participants after the experiment in the three stimulating environments: (**a**) Systolic blood pressure change; (**b**) Diastolic blood pressure; (**c**) Peripheral oxygen saturation change; (**d**) Heart rate change. (*n* = 120; mean ± SD; * *p* < 0.05; ** *p* < 0.01; verified by a paired t-test, DFIS: deciduous forest interior setting; DFTS: deciduous forest trail setting; CS: city setting).

**Figure 5 ijerph-19-03453-f005:**
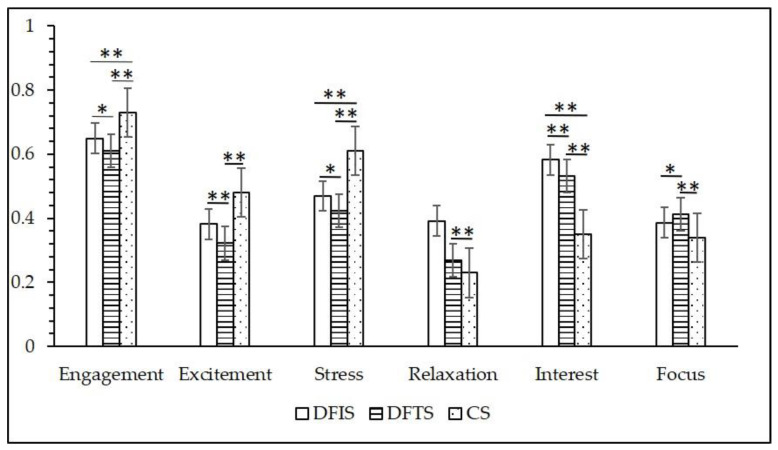
Neural emotional parameter values for the three stimulating environments (*n* = 120; mean ± SD; * *p* < 0.05; ** *p* < 0.01; verified by a paired *t* test, DFIS: deciduous forest interior setting; DFTS: deciduous forest trail setting; CS: city setting).

**Figure 6 ijerph-19-03453-f006:**
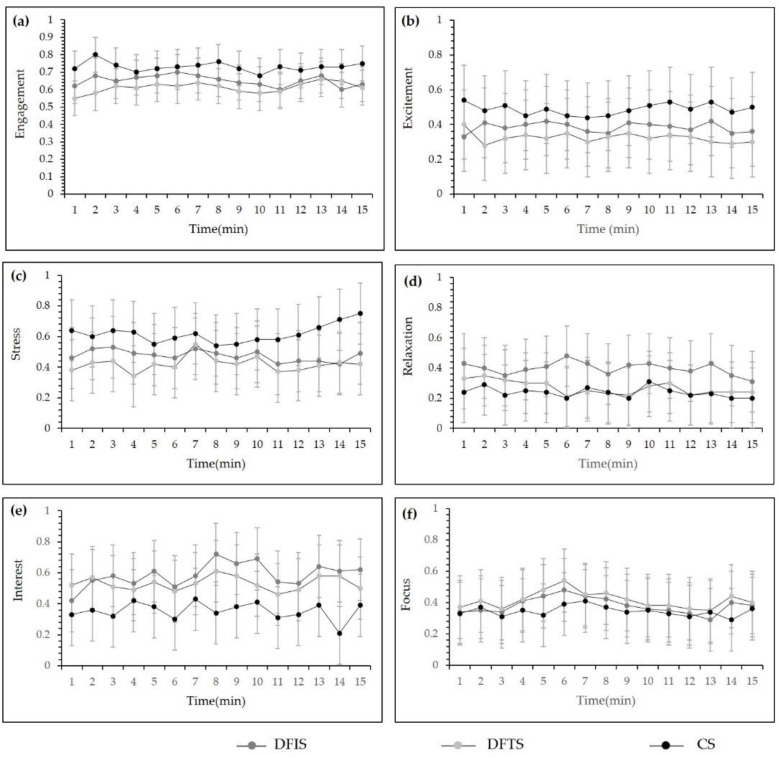
Minute-by-minute dynamic values of the neural emotional parameters for the three stimulating environments: (**a**) “Engagement” emotional parameters dynamic values; (**b**) “Excitement” emotional parameters dynamic values; (**c**) “Stress” emotional parameters dynamic values; (**d**) “Relaxation” emotional parameters dynamic values; (**e**) “Interest” emotional parameters dynamic values; (**f**) “Focus” emotional parameters dynamic values;(*n* = 120; mean ± SD; DFIS: deciduous forest interior setting; DFTS: deciduous forest trail setting; CS: city setting).

**Figure 7 ijerph-19-03453-f007:**
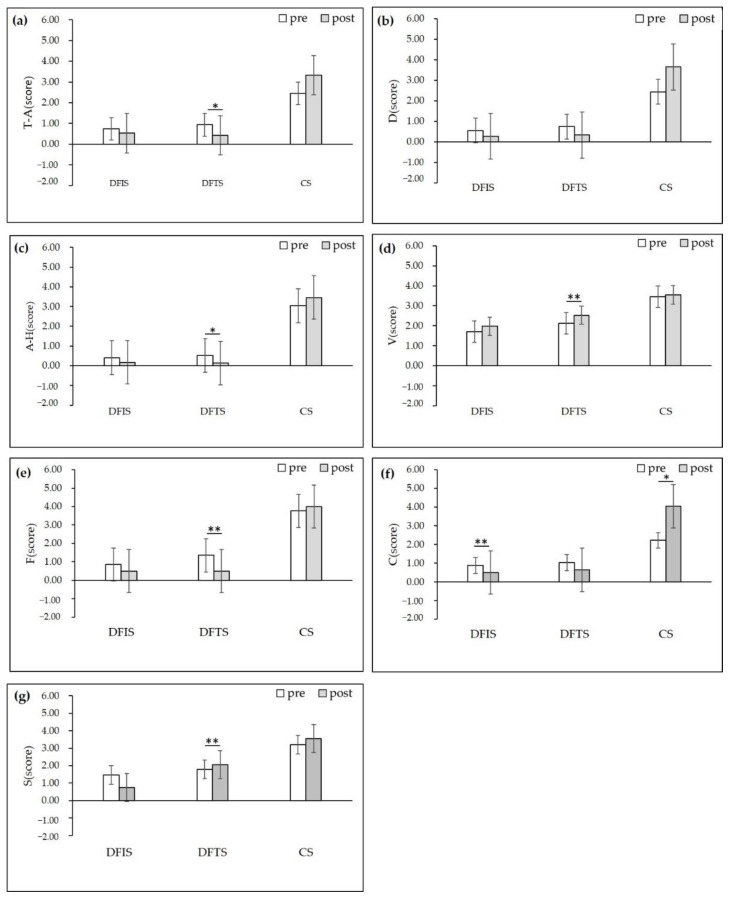
POMS scores before and after the three stimulating environments: (**a**) T-A scores change; T-A: tension-anxiety; (**b**) D scores change; D: depression; (**c**) A-H scores change; A-H: anger-hostility (**d**) V scores change; V: vigor; (**e**) F scores change; F: fatigue; (**f**) C scores change; C: confusion; (**g**) S scores change; S: self-esteem; (*n* = 120; mean ± SD; * *p* < 0.05; ** *p* < 0.01, verified by Wilcoxon signed-rank test; DFIS: deciduous forest interior setting; DFTS: deciduous forest trail setting; CS: city setting).

**Figure 8 ijerph-19-03453-f008:**
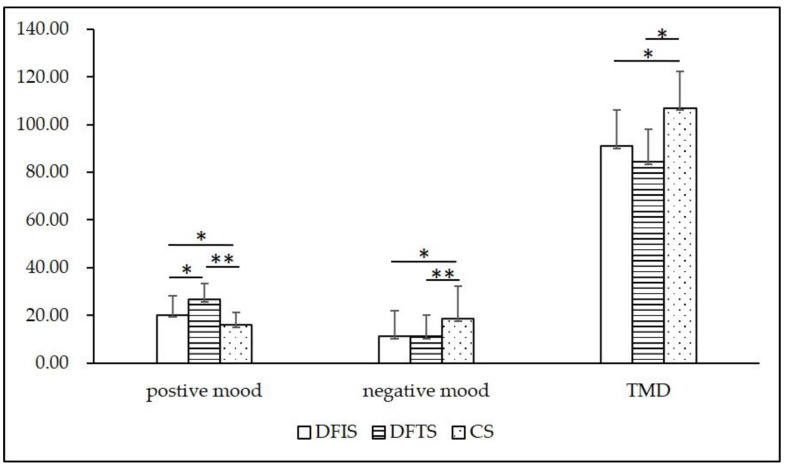
Mood state values for the three stimulating environments (*n* = 120; mean ± SD; * *p* < 0.05; ** *p* < 0.01; verified by Wilcoxon signed-rank test; DFIS: deciduous forest interior setting; DFTS: deciduous forest trail setting; CS: city setting; “TMD” means “total mood”).

**Table 1 ijerph-19-03453-t001:** Environmental factors of the three sites.

Parameter (Mean ± SD)	3 November 2020			4 November 2020			5 November 2020		
	DFIS	DFTS	CS	DFIS	DFTS	CS	DFIS	DFTS	CS
Temperature (°C)	16.38 ± 1.32	16.25 ± 1.25	17.11 ± 1.34	15.48 ± 1.07	15.33 ± 0.75	15.86 ± 0.84	14.72 ± 1.30	14.48 ± 1.12	15.66 ± 1.20
Humidity (%)	69.51 ± 3.17	69.02 ± 3.26	58.38 ± 1.23	74.90 ± 3.15	72.8 ± 4.8	68.55 ± 4.31	84.10 ± 6.57	85.21 ± 5.47	82.44 ± 3.25
Noise	51.54 ± 17.45	62.39 ± 7.25	88.53 ± 12.76	56.50 ± 17.36	59.91 ± 10.48	84.52 ± 6.72	61.1 ± 7.28	64.38 ± 5.82	77.93 ± 7.28
Light intensity	90.41 ± 64.31	95.83 ± 74.71	91.26 ± 61.93	106.58 ± 31.00	136.74 ± 44.23	148.17 ± 52.36	107.36 ± 44.23	116.89 ± 41.40	147.20 ± 56.22

Data are presented as the mean ± SD; DFIS: deciduous forest interior setting; DFTS: deciduous forest trail setting; CS: city setting.

**Table 2 ijerph-19-03453-t002:** Environmental preference and spatial scale for deciduous landscape type.

Serial Number	Evaluation Indicator	Notes	Scores
1	Environmental preference	Preference for environment	0 1 2 3 4
2	Density	Aggregation and density of plants	0 1 2 3 4
3	Height	Perception in vertical view and plant height	0 1 2 3 4
4	Level	Perception in horizontal view and plant width	0 1 2 3 4
5	Overall	Overall feeling regarding entire environment within the scope of vision	0 1 2 3 4

**Table 3 ijerph-19-03453-t003:** Effects of the three stimulating environments on restorative states.

Rrestorative States Associated with the DFIS							
ROS	Pretest		Posttest		z	*p*	Change Rate
	Mean	S.D.	Mean	S.D.			
ROS scores	4.39	1.04	5.03	1.05	−2.53	0.01 *	0.64
Restorative states associated with the DFTS							
ROS	Pretest		Posttest		z	*p*	Change rate
	Mean	S.D.	Mean	S.D.			
ROS scores	4.6	1	5.31	0.87	−3.1	0.00 **	0.71
Restorative states associated with the CS							
ROS	Pretest		Posttest		z	*p*	Change rate
	Mean	S.D.	Mean	S.D.			
ROS scores	3.39	0.9	3.14	0.79	−2.55	0.06	−0.25

(*n* = 120; mean ± SD; * *p* < 0.05; ** *p* < 0.01; verified by Wilcoxon signed-rank test; ROS: Restorative Outcomes Scale; DFIS: deciduous forest interior setting; DFTS: deciduous forest trail setting; CS: city setting).

**Table 4 ijerph-19-03453-t004:** Pearson’s coefficients for the correlations between the physiological/psychological indicators and the deciduous landscape space elements.

Physiological and Psychological Indicators	Preference	Density	Height	Level	Overall
Systolic blood pressure	−0.23	−0.02	−0.13	−0.01	0.13
Diastolic blood pressure	−0.16	0.03	−0.01	0.02	0.28
HR	0.15	−0.26	0.04	−0.30	−0.03
Blood oxygen	−0.05	0.08	−0.26	−0.01	−0.34
Positive mood	0.54 **	0.04	−0.02	0.30	0.01 *
Negative mood	−0.29	0.04	0.14	0.01	0.04
TMD	0.48 **	0.07	0.17	−0.13	0.00
ROS	0.17	0.01	−0.23	0.10	−0.11
Engagement	−0.02	0.15	0.22	0.56 **	0.19
Excitemrnt	0.04	0.10	0.16	0.11	−0.05
Stress	−0.11	−0.28	0.09	0.06	−0.18
Relaxation	−0.29	−0.41 **	−0.37 *	−0.43 **	−0.18
Interest	−0.18	−0.39 *	−0.13	−0.41 **	−0.09
Focus	−0.17	0.31	0.31	−0.31 *	−0.17

* *p* < 0.05; ** *p* < 0.01.

## Data Availability

Not applicable.

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
