# Peer review of "An Empirical Study of the Restoration Potential of Urban Deciduous Forest Space to Youth"

_ijerph, 2022, doi:10.3390/ijerph19063453_

Round 1
Reviewer 1 Report
The authors presented an interesting study on the restoration potential of urban deciduous forest space to youth. The structure of this paper is fine, but English needs polishing. The authors are encouraged to conduct more appropriate writings to improve this manuscript. This reviewer has some comments regarding the clarity of the paper:
-Line 12-36: It is recommended to shorten the words and focus on interesting results in Abstract.
-Line 37: “urban forest” vs “urban forests” Please consider to use a unified expression of “urban forests” throughout this manuscript.
-Line 41-48:
1)Please consider directly describing the progress of urban forest research in the first paragraph of the introduction.
2) Authors mentions “possible biological mechanisms, elemental characteristics and the health and well-being of residents.. ” . It is lacking for key literatures. Perhaps some recent refs are, for example: 1) Hong, X. C., Wang, G. Y., Liu, J., et al. (2021) Modeling the impact of soundscape drivers on perceived birdsongs in urban forests. Journal of Cleaner Production, 292, 125315; 2) Hao, Z. , Wang, C. , Sun, Z. , Bosch, C. , & Pei, N. . (2020). Soundscape mapping for spatial-temporal estimate on bird activities in urban forests. Urban Forestry & Urban Greening, 126822. 3) Lin B. B., Ossola A., Alberti M., et al. (2021) Integrating solutions to adapt cities for climate change. The Lancet Planetary Health 5(7):e479-e486.
-Line 148: Figure 1. Please consider to simplify location maps, and adding site details.
-Line 151-170 Perhaps describe the trees, shrubs and groundcover plants in the study area, including species names, characteristics, etc.
-Other comments in Discussion:
1)Line 339-340: “It is known that... ”It is lacking for literatures.
2)Some parts are weakly connected logic to follow, especially 4.1 to 4.5 part.
3)Perhaps more details for plant planning.
Reviewer 2 Report
A couple of minor corrections that are noted on the attached fiel

Reviewer 3 Report
The manuscript titled “An empirical study on the restoration potential of urban deciduous forest space to youth” statistically analyzed physiological and psychological effects of urban spaces with deciduous tree species to youth in the urban setting.
I would suggest this manuscript returned to the authors for minor revisions to adequately address the concerns/suggestions pointed out above and detailed below.
Structural concerns/suggestions:
- Lines 137-146: I would suggest to re-construct this paragraph to better organize the information for the three study sites.
- Based on the expression, “final” (line 144), I guess that there were some candidate sites for DFIS but somehow (with some selection criteria??) the “area of 84×72 m was used as the environmental stimulation site for DFIS”. If this is the case, addressing more information for the selection would be better. In addition, these two sentences may be mentioned together.
- I guess that “The road with ginkgo biloba on both sides of the DFTS test site was selected, and the road width was 1.5-3 m.” explains the candidate sites and “The DFTS test site was a 166×3.2 m U-shaped trail” explains the actual test site selected somehow. These two sentences can be mentioned together but more explanations between these two sentences may be needed.
- In addition, if my guess above is correct, then the sentence “The road with ginkgo biloba on both sides of the DFTS test site was selected, and the road width was 1.5-3 m.” can be modified, such as “The roads (1.5-3 m width) with ginkgo biloba on both sides were selected as candidates for the DFTS test site.”
- Line 144: Does “T-turn” mean “T-intersection”? “T-intersection” is more generally used.
- The expression for the control site may be “The control sample site was a 46×24 m block open space at an urban T-intersection.”
- Below is the sectional structure (sections 2-4) of the manuscript.
- Please re-consider the section titles and structure to better represent the correspondence among method, results, and discussions. For instance, correspondence between 2.3.2 Psychological measurement (this should be Physiological measurement, though) and 3.1. Physiological results are well corresponding, but the term “Neuroemotional” appears for the first time for the section title in 3.1.2. To better present the correspondence, this term should at least appear in the text in section 2.3.2. Same for “Restorative emotions” used for the section title in 3.2.2; this term was never used previously in the method section.
- Subsections in the section 4 are somewhat corresponding to those in previous sections but can the be shorter?
|
2. Materials and Methods |
|
2.1. Participants |
|
2.2. Study Sites and Procedures |
|
2.3. Measure |
|
2.3.1. Sociodemographic characteristics |
|
2.3.2. Psychological measurement |
|
2.3.3. Psychometrics |
|
2.3.4. Preference and spatial scale score |
|
2.4. Statistical analysis |
|
3. Results |
|
3.1. Physiological results |
|
3.1.1. Blood oxygen, blood pressure, and pulse rate |
|
3.1.2. Neuroemotional parameters |
|
3.2. Psychometric results |
|
3.2.1. POMS Mood |
|
3.2.2. Restorative emotions |
|
3.2.3. Preferences for the deciduous environment |
|
4. Discussion |
|
4.1. Empirical evidence of the restoration potential of defoliated landscape spaces for youth |
|
4.2. Effects of different deciduous forest spaces on blood pressure, blood oxygen, and pulse rate in adolescents |
|
4.3. Effects of different deciduous forest spaces on adolescent EEG |
|
4.4. Psychological effects of different deciduous forest spaces on adolescents |
|
4.5. Spatial scale preference of adolescents for deciduous forests |
|
4.6. Proposal for urban construction planning based on recovery potential |
Expressional concerns /suggestions:
- Lines 65-67: The sentence, “nontree data based on positive health impacts need to be valued in public health as health measures for innovative funding streams or to guide forestry strategies.”, is not intuitively understandable. I’d suggest the authors to rephrase it to better convey what they meant to say.
- Line 99: “Chengdu” appeared here for the first time without any explanation. By going forward up to line 138 I understood it meant the name of the study city but it would be more reader-friendly to change it at least “Chengdu, Sichuan Province”.
- Line 161: It may be better to spell out “EEG” when it appears for the first time in the manuscript. Then electroencephalogram in line 186 can be replaced with EEG.
- Table 1: It may better organize the information in this table by dividing each column (e.g., 2020.11.3) into three columns for DFIS, DFTS, and CS. For instance, see below.
|
Parameter (Mean±SD) |
2020.11.3 |
2020.11.4 |
2020.11.5 |
||||||
|
DFIS |
DFTS |
CS |
DFIS |
DFTS |
CS |
DFIS |
DFTS |
CS |
|
|
|
|
|
|
|
|
|
|
|
|
- Line 204: This is only caption for Figure 3. No figure is there.
- Table 2: It is just an aesthetic suggestion, but the table may be better with the one shown below. i.e., the second and third columns aligned left rather than center. Wording consistency for the notes for height and level.
|
Serial number |
Evaluating indicator |
Notes |
Scores |
||||
|
1 |
Environmental preference |
Preference for environment |
0 |
1 |
2 |
3 |
4 |
|
2 |
Density |
The aggregation and density of plants |
0 |
1 |
2 |
3 |
4 |
|
3 |
Height |
Perception in vertical view and plant height |
0 |
1 |
2 |
3 |
4 |
|
4 |
Level |
Perception in horizontal view and plant width |
0 |
1 |
2 |
3 |
4 |
|
5 |
Overall |
Overall feeling of whole environment within the scope of vision |
0 |
1 |
2 |
3 |
4 |
- items in “evaluating indicator” and “notes” may be better to be aligned left rather than center. For instance, “Environmental preference”
- Figures 4, 6, and 7: I would suggest adding (a), (b), (c), …, etc. to these figures as you did in Figure 1, and put explanations for each panel in the figure caption.
- Line 241: The text says “p<0.01”, whereas there is only one asterisk (*) in Figure 4 HR figure (that represents p<0.05). Which is correct?
- Line 259: The order of “interest” and “excitement” in the text may be better with the same order (left to right, i.e., “excitement” and “interest”) in Figure 5. That way, readers may feel easier to follow.
- Lines 258-259: Is this statement correct? In Figure 6, DFIS is higher for “excitement” and “interest”.
- Line 283: “mainly,” sounds a bit odd because there are only two positive emotions, and you refer these two. If there are, for instance five positive emotions, and only two of them increased then it sounds more natural to say “mainly” but this is not the case. You can delete it.
- Lines 287-291: You used “M” and “TMD” without any explanations. It can be guessed that “M” means “mean” and “TMD” means total mood but it should be clearly defined.
- Figure 7: What is “1” in “1pre” and “1post” legend in the upper right corner of each panel? Also, “D 1pre” and “D 1post” are used in the top-right panel.
- Figure 7: Explanation of the “S: Self-esteem” is missing.
- Figure 8: what is “TMD”?
- Lines 342-347: This whole sentence seems quite long. You may consider reconstruct it (i.e., shorten or separate to multiple sentences).
- Lines 365-367: I couldn’t get the meaning of this sentence. Please rephrase it.
- Line 371-374: These two sentences are confusing. The first sentence starting from “but the DFIS” means DFIS<DFTS, while the second sentence denotes that the hypothesis was DFIS<DFTS. They are not inconsistent. Your original hypothesis is “(2) DFIS improves participants' physical and mental states more significantly (line 113)”, meaning DFIS>DFTS. Thus, your second sentence was erroneous.
- I would suggest going through the expression in References to make sure they are correct. There are obvious errors like “67 Hiroko, O.; Harumi, I, …, Maiko, K.; Takashi, M.; Takahide, K.;…” those are all first names that spelled and only the initial for the last name is presented. It should be opposite.
- Also, please make sure the correspondence between the text and reference list.
- Discussion section: Throughout the section, it would improve the readability if you put table and figure numbers in the text where they are referred to. For instance, lines 469-470 can be “We found that the indicators of relaxation and interest decreased significantly 469 when density and level increased to a certain level (Table 4)”.
- Line 405: If you have a particular meaning in the order of “engagement”, “interest” and “stress” then that’s fine but otherwise the order should be “engagement”, “stress” and “interest” that is consistent with the order in Fig.5. That’s easier for readers to follow. Please go through the manuscript to make sure this kind of the improvement of the readability.
- Line 408: It says, “three parameters” but it seems that only “Engagement” and “Stress” were significantly lower than the control group.
- Line 415: “Second” appears without its correspondence, i.e., “First”. It would be easier to follow you put “First” in the paragraph that denotes the first discussion in this section.
- Lines 426-462: It would make more sense to put “however” between the two sentences, i.e., “we expected the DFIS with more natural elements to stimulate a higher preference; however, the DFTS group was more popular.”
- “youth” and “adolescent” are interchangeably used throughout the manuscript but it may be better to consistently use one term.
Minor concerns/suggestions (typo, clarity, wording, grammar, tense, etc.)
- Line 77: “ompared” should be “compared”.
- Line 77: “The overall” should be “the overall”.
- Throughout the manuscript, words “restoration”, “restorative”, “restored” are used. Below are some examples of this:
- Line 14: restorative effects
- Line 34: restorative environments
- Line 48: restored environments
- Line 50: spatial restoration effects
- Line 80: seasonal green space restoration research
It may be obvious that these expressions of “restore” mean the improvement of human well-beings in the context of physiological and/or psychological studies. But, on the other hand, this expression is often used to represent the “restoration” of urban spaces. In fact, the examples c, d, and e may be confusing as they could be interpreted as “restoration” of spaces. If you used “physiological/psychological restoration” or something like that in the beginning of the introduction section, it could be much clearer what “restore” means throughout the manuscript.
- Line 109: What does “200 mu” mean?
- Line 183: Should “Psychological” be “Physiological” in the section title?
- Line 213: A Chinese character “是” should be removed.
- Line 254: “indexes” should be “indices”.
- Line 291: Is “M=” missing after “DFTS:”?
- Line 327: “significant negatively” should be “significantly negatively”.
- Table 4: Is “*” missing for overall-Positive mood (0.01)? The text in line 332 denotes “"overall" was significantly positively correlated with "positive mood" (0.005, p < 0.05)”.
- Table 4: “Postive mood” should be “Positive mood”.
- Line 365: “so” should be “So”.
- Line 389: What is “ja, B”?
- Line 390: Lee, J [67] may not be the correct format for citation. It does not point the correct citation.
- Line 391: Deng, L. et al. [12] does not point the correct citation.
- Lines 509, 510, 513: a hyphen (“-“) should be removed from “diver-sity”, “ele-ments”, and “investiga-tion” I am afraid that the sentences including these errors might have been copied from pdf files or something like that and pasted here. If that’s the case please be aware that it might be considered as plagiarism.
Reviewer 4 Report
Relevance:
The manuscript is relevant for the field and presented in a well-structured manner. The hypothesis is clear and the case study selected are appropriate to test it.
Areas of weakness to be reviewed or complemented:
Several identified aspects appear in the results discussion section. However, from a global perspective, a discussion would be necessary to explore the role of these results in two specific current situations: on the one hand regarding the pandemic and post-pandemic context. On the other hand regarding the climate change crisis context.
With regards to these suggestion, the following reference may help in this complementary but necessary current approach of the manuscript:
- Green space and mortality in European cities: a health impact assessment study (2021)
DOI:https://doi.org/10.1016/S2542-5196(21)00229-1
The capacity of access of the youth to the selected spaces is also an issue that could be debated and questioned. In this sense, the concept of Just City could complement the discussion of results (see https://spatialjustice.blog/susan-fainstein-and-the-just-city/). The role of landscapers and urban designers from a more intersectional approach also opens a field on which recommendations could be developed, and event the role of youth in decision-making and community-driven urban design.
Specific comments:
The introduction of images (pictures) of the different deciduous forests to be compared might provide in a quicker understanding and perception of the specificity and atmosphere of each case.
Round 2
Reviewer 1 Report
Very unique research done.
All my comments have been clarified in this Revision.
I recommend this publication.
Author Response
Thank you for your support and encouragement to our research. We are glad that this revision has met your expectations. Thank you again for your valuable time and constructive suggestions.
Best wishes!
Reviewer 3 Report
The manuscript was well revised according to suggestions made by the reviewer, but there are still some inconsistencies and errors. I would suggest this manuscript returned to the authors for minor revisions.
- Line 145: The unit “mu” is a Chinese unit. Please use a SI unit (i.e., m2 for area).
- Line 147: “20 minute' drive” should be “20 minutes drive” or “20-minute drive”.
- Line 148: Is there any unit for “0.45”?
- Line 152: Please remove “s, “ in “trails, s, i.e.,”
- Line 160: “m” should be “m2”
- Line 177: “EEG” first appeared here and then “the electroencephalogram (EEG)” appeared at line 179. It should be spelled out at line 177.
- Table 1: It is just a suggestion, but it may be aesthetically better to have numbers fit in a single row rather than two rows. You can just use a smaller font for this.
- Line 253: “indexes” should be “indices”
- Lines 266-268: These two sentences can be combined to a single sentence as they both describe the indicators where the DFIS is higher than the DFTS.
- Line 270: "relaxation" should be “Relaxation” and a single quotation (“) after “relaxation” should be removed.
- Caption for Figure 7: “(a)T-A scores change” and “T-A: tension-anxiety” can be combined. Other items too.
- Lines 369-370: The results show DFIS < DFTS. Your second hypothesis is DFIS > DFTS. So, it is true that they are inconsistent. But your sentence, “our second hypothesis that the deciduous forest trail space would more significantly improve the physiological status of the participants” is still erroneous. See lines 118-119 for the second hypothesis “(2) the DFIS will improves participants' physical and mental states more significantly” Please correct.
- Line 656: The citation is still incorrect. Hiroko, Harumi, Maiko, Takashi, Takahide are first names. Usually, last names are spelled out, and only an initial for first names is used.
Author Response
We would like to thank you for careful and thorough reading of this manuscript again, which help to improve the quality of this manuscript.The attachment is a point-by-point response to your comments and concerns.Thank you again for your careful and helpful review comments!
Best Wishes.
